# Rapid System to Detect Variants of SARS-CoV-2 in Nasopharyngeal Swabs

**DOI:** 10.3390/v15020353

**Published:** 2023-01-26

**Authors:** Marco Favaro, Paola Zampini, Enrico Salvatore Pistoia, Roberta Gaziano, Sandro Grelli, Carla Fontana

**Affiliations:** 1Department Experimental Medicine, University of Rome “Tor Vergata”, 00133 Rome, Italy; favaro@uniroma2.it (M.F.); pistoiae@uniroma2.it (E.S.P.); roberta.gaziano@uniroma2.it (R.G.); grelli@med.uniroma2.it (S.G.); 2Via Luigi Einaudi, Guidonia Montecelio, 00012 Rome, Italy; paola.zampini@adaltis.net; 3National Institute for Infectious Diseases Lazzaro Spallanzani IRCCS, 00149 Rome, Italy

**Keywords:** SARS-CoV-2, variants, rapid system

## Abstract

Currently, the reference method for identifying the presence of variants of SARS-CoV-2 is whole genome sequencing. Although it is less expensive than in the past, it is still time-consuming, and interpreting the results is difficult, requiring staff with specific skills who are not always available in diagnostic laboratories. The test presented in this study aimed to detect, using traditional real-time PCR, the presence of the main variants described for the spike protein of the SARS-CoV-2 genome. The primers and probes were designed to detect the main deletions that characterize the different variants. The amplification targets were deletions in the S gene: 25–27, 69–70, 241–243, and 157–158. In the *ORF1a* gene, the deletion 3675–3677 was chosen. Some of these mutations can be considered specific variants, while others can be identified by the simultaneous presence of one or more deletions. We avoided using point mutations in order to improve the speed of the test. Our test can help clinical and medical microbiologists quickly recognize the presence of variants in biological samples (particularly nasopharyngeal swabs). The test can also be used to identify variants of the virus that could potentially be more diffusive as well as not responsive to the vaccine.

## 1. Introduction

Since the appearance of SARS-CoV-2 in late 2019, many people worldwide have presented with severe pneumonia at hospitals [1]. Over time, the number of patients rapidly increased. Community transmission of the virus, as well as anti-viral treatments, can promote mutations in the virus, resulting in a more virulent and diffusive virus with a potentially higher mortality rate [2,3]. Indeed, another aspect that can act on virus evolution is the mutation rate, which is a function of accuracy in replication; the latter also represents the intrinsic rate that guides the genetic changes upon which selection can act [4].

Even though most of the emerging mutations do not have a substantial impact on the spread of the virus or on its virulence, many others can provide selective advantages, which can be summarized as increased transmissibility, the ability to escape from the host’s immune response, resistance to anti-viral drugs, and vaccine effectiveness [2,5,6,7,8]. The world has faced five main variants of SARS-CoV-2 defined as variants of concern (VOCs): variant B.1.17 from the United Kingdom, variant P.1 from Brazil and Japan, variant B1.617.2 from India, and variants B1.351 and B.1.1.529 from South Africa. In May 2021, the WHO proposed an easier way to identify these variants: Alpha, Gamma, Delta, Beta, and Omicron [5,6,7,8,9,10].

All these variants likely have not had an impact on the mortality rate but led to increased transmissibility, especially for Gamma and Beta, in which point mutations K417N and E484K could weaken the effectiveness of vaccines, all of which resulted in a worsening of the epidemiological situation across the globe [11,12].

A worrying aspect concerns Omicron; this variant has numerous mutations and deletions, some of which are also present in other VOCs [13,14]. Moreover, Omicron can evade humoral immune protection developed after vaccination [15]. Over time, several Omicron subvariants have been identified and divided into three main sublineages, BA.1 (B.1.1.529.1), BA.2 (B.1.1.529.2), and BA.3 (B.1.1.529.3) [16]. Recently, BA.4 and BA.5, lineages that both contain the amino acid substitutions L452R, F486V, and R493Q in the spike receptor-binding domain compared with BA.2, have been added to the list of variants [17]. Presently, BA.4/5 lineage subvariants represent approximately 77.1% of all Omicron-related lineages. In addition, BQ.1 and BF.7, which are two of many BA.4/5 subvariants, are worrisome because they show increased immune evasion as well as resistance to monoclonal antibodies. Furthermore, BF.7 contains R346Twhile BQ.1, defined by ECDC as a variant of interest, shows both K444T and N460K spike mutations [18,19].

To identify and trace these variants, researchers have been using whole genome sequencing (WGS) [20,21,22,23,24]. WGS, although it represents the standard, is a time-consuming and expensive technique that is mainly available in large laboratories or in national reference laboratories. On the other hand, WGS has the great advantage of helping in defining emerging variants that can be lineage-specific and can be used for proper identification [25,26]. Nevertheless, for the early identification of such variants, it is desirable to use a molecular system that is both easy to use and affordable [27]. Additionally, the National Institute of Health published a framework to detect Omicron BA.4/BA.5 subvariants through real-time PCR. The reported PCR assays that target specific mutations can be a useful tool for the timely detection of variants/subvariants, including Omicron BA.4/BA.5 [28,29,30].

Here, we present a TaqMan-based real-time PCR designed to characterize SARS-CoV-2 variants. The real-time PCR assay uses a master mix to simultaneously detect the main deletions associated with the variants described above, located in the spike protein and *ORF1a* genes. Our assay employed six sets of primers and probes for the identification of variants, and one set dedicated to the amplification of the beta-actin gene was used as an amplification control.

## 2. Materials and Methods

### 2.1. Samples

This study did not include human participants but, rather, leftover samples. For our assay, we used nucleic acids (NCs) that had already been extracted from 400 nasopharyngeal swabs (NPSs) routinely processed using the Nimbus instrument (Seegene, Inc., Seoul, Republic of Korea). NPSs were delivered to the microbiology laboratory of our hospital from March 2021 to March 2022. Any positive NPSs were determined to be positive in the base results obtained by using a commercial system, namely, Allplex TM 2019-nCov Assay-Seegene, and by the method described in our previous work [27]. The NCs included in our study were randomly selected among positive NPSs whose CTs were ≤38. NCs were stored at −80 °C before testing. To ensure the RNA integrity of the stored NCs, they were retested by our assay [20]. The criteria for inclusion of NCs were the presence of the same genes detected at the time of the first amplification as well as a comparable number of CTs (those of the first amplification ± 2). NCs for which the presence of the genes detected in the first evaluation was not confirmed were excluded. Five μL of the eluate was used for our assay.

As a positive control in our assay, we used commercial samples designed to detect the Alpha, Beta, Gamma, and Delta variants from Twist Bioscience, namely, Twist Synthetic SARS-CoV-2 RNA Controls 15, 16, 17, and 18 for Alpha, Beta, Gamma, and Delta identification, respectively (Twist Bioscience, San Francisco, CA, USA).

### 2.2. Primers and Probes

For the identification of the specific deletions, our multiplex real-time PCR assay used six sets of primers and probes. Although the test was performed on known positive samples, an internal control was nevertheless included (beta-actin gene). Table 1 reports the primer and probe sequences as well as labeling fluorophores for each probe. Primers and probes were both synthesized by Metabion International AG (Metabion, Planegg, Germany) and by Bio-Fab (Rome, Italy).

### 2.3. PCR Conditions

The working solution contained a mix of six primers and probes in a single tube, and the final concentration for each reaction is shown in Table 1. Taq DNA Polymerase and Reverse Transcriptase qPCRBIO Probe 1-Step Go No-ROX (cat. PB2543) were used according to the manufacturer’s instructions (PCR BIOSYSTEMS, PCR Biosystems Ltd., London, UK). Real-time PCR conditions were as follows: reverse transcription for 10 min at 45 °C, RT inactivation/Taq DNA polymerase activation for 2 min at 95 °C, followed by 40 cycles of 15 s at 95 °C and 30 s at 60 °C. Real-time PCR tests were run on an Amplilab real-time machine (Adaltis srl, Guidonia Montecelio, Italy). The results of our multiplex PCR assay are shown in Table 2. The specificity of our assay was evaluated by performing a PCR test in which the working solution was tested on a mix containing all the positive controls: Twist Synthetic SARS-CoV-2 RNA Controls 15, 16, 17, and 18 (Twist Bioscience) and an aliquot of a positive clinical sample known to be the Omicron variant. The sensitivity of our assay was explored by using a 10-fold serial dilution of each control from Twist Bioscience as well as an Omicron positive sample.

### 2.4. Sequence Analysis

To confirm the presence of the deletions in the products obtained from our multiplex PCR assay, amplicons from a separate amplification of a large part of the S gene (containing all deletions) were sequenced by the Sanger method using either the sequencing service by Bio-Fab research (Bio-Fab research, Rome, Italy) or by in-house sequencing performed using BigDye Terminators V1.1 (Applied Biosystems, Foster City, CA, USA) and analyzed with ABI Prism 310 (Applied Biosystems). The following primers were used: S seq F 5′ CCA CTA GTC TCT AGT CAG TGT GT 3′ and S seq R 5′ GAG AGG GTC AAG TGC ACA GT 3′ (this work).

All methods described were carried out in accordance with relevant guidelines and regulations.

## 3. Results

The results of our assay performed on commercial samples provided by Twist Bioscience were in total agreement with those reported in the IFU. At the time of our study, a sample for the Omicron variant was not yet available, so to confirm the performance of our method in characterizing this variant, we used a sample known to be classified as Omicron (the sample was processed by NGS and belonged to a set of specimens of SARS-CoV-2 processed in a national survey of SARS-CoV-2).

From a total of 400 NCs processed by our assay, we obtained the following results: 100 NCs (100/400; 25%) were positive for Alpha, 200 were positive for Omicron (200/400, 50%), eight (8/400; 2%) were positive for Beta, and four (4/400; 1%) were positive for Delta. Gamma was not detected in the specimens analyzed. The remaining samples were identified as SARS-CoV-2 strains (wild type) and/or as variants other than Alpha, Beta, Gamma, Delta, or Omicron.

To confirm the nature of the variants identified by our assay, samples were analyzed by sequencing. An overall agreement of 98% was recorded between the sequencing and PCR results. In particular, 100 samples showed a sequence compatible with deletions 69–70 (Alpha variant), eight samples were confirmed as Beta and four as Delta. Two hundred samples, without deletions 69–70 and showing the FAM signal, were classified as positive for the Omicron variant. Mixed electropherograms with overlapping peaks were observed in ten samples. Finally, the remaining 78 samples did not show deletions or relevant mutations when compared with the wild-type strain. Figure 1 reports sequences of a variant with a 69/70 and 3675/3677 deletion, while Figure 2 shows a mixed electropherogram. The latter was likely due to the presence of two viral genomes simultaneously in the same samples. The possible presence of two different lineages was also reported by Benites et al. and by Richard L Tillett et al. [3,30]. Table 2 reports the interpretation of the results of our assay. It must be considered that the presence of the variant was ascertained by comparing the signals registered in the different channels (FAM, ROX, Cy5, Cy5.5) after the amplification assay. In fact, the Alpha, Beta, and Gamma variants have the deletion on *ORF1a* in common, while deletions 69–70 are only present in the Alpha variant, and deletion 25/27 is present in Omicron. Deletion 241/243 characterizes the Beta variant. Thus, if in our assay there were signals for two fluorophores (TX RED, CY5.5) (identifying deletion 241/243 as well as 3675–3677), it would mean that the Beta variant was present. Otherwise, in the presence of a unique signal for the fluorophore (Cy5) (157/158 deletion), we can conclude that the sample is positive for the Delta variant. A combination other than those described may indicate a new mutation or a coinfection of several variants. The sensitivity and specificity of our test are reported in Figure 3 and Figure 4, respectively. No cross-reaction has been recorded (Figure 3) while the sensitivity is equal to 10 reaction copies (Figure 4).

## 4. Discussion

VOC 202012/01 (known as Alpha) was the first variant identified in the southern United Kingdom in December 2020, although it traced back to September 2020 [14]. It rapidly became the predominant variant in circulation in the UK and became a variant of concern due to its increased transmissibility [31,32]. The UK implemented stricter non-pharmaceutical interventions (NPIs) to reduce transmission [14]. Additionally, in Denmark, community transmission of VOC 202012/01 was observed, and in response, the country strengthened and prolonged containment measures. In December 2020, the variant 501Y.V2 (now Beta) was first identified in South Africa and quickly became one of the most prevalent variants. Additionally, this variant was characterized by increased transmissibility, and starting in January 2021, it was identified in ten EU/EEA countries (starting with France) as well as in Israel and the UK [33]. Starting in December 2020, the SARS-CoV-2 lineages B.1.617.1, B.1.617.2, and B.1.617.3 (also known as Delta) were first reported in India, and then the same variants were increasingly detected in many other countries [13]. This is the variant that quickly spread around the world until the arrival of the newest isolated variant once again in South Africa, known today as Omicron. The world is facing a rapid diffusion of new variants and subvariants (BA.4, BA.5, BQ.1, and BF.7) [15,19]. On the basis of previous experiences, the increasing proportion of new variants/subvariants might have a substantial impact on transmissibility, severity and/or immunity, which likely has an effect on the epidemiological situation in the EU/EEA [19,29].

Therefore, it appears clear that to contain the spread of such variants, their timely identification is extremely urgent and necessary. The reference method to identify variants of SARS-CoV-2 is whole genome sequencing; however, as stated above, it is expensive, time-consuming, and limited to large and reference laboratories only. Our system was proven to be helpful since the detection of the main variant of the virus was faster and more affordable. The method is a simple real-time PCR assay, and it does not require expensive instrumentation. Our assay is easy to use and can be introduced in every laboratory, even in those that may not have advanced sequencing systems available. Its advantage is that every hospital will be able to quickly detect/identify variant circulation to promptly implement the infection control measures required to prevent further transmission in their setting [34,35].

Our assay helped us to quickly confirm the presence of different variants among our specimens and to exclude others. Since our test is based on the direct detection of the presence of deletions, it is not affected by the possible co-presence of wild-type SARS-CoV-2 in the specimens (which indicates a coinfection). As a matter of fact, some of the diagnostic tests commercially available base their detection of variants on the absence of S gene amplification; this design fails in the presence of a coinfection or a reinfection (a condition that may generate the co-presence of two types of viruses in the same sample), and the test could result in false negatives. A possible limitation of our paper is that among the samples included in this study, no Gamma variant was detected. However, starting from the evidence that our assay correctly identified Gamma, using a commercially available specific positive control, we can speculate that our real-time PCR can also work when used on NSF specimens.

## 5. Conclusions

The results of our assay allow us to have a concrete idea of the real circulation of variants in our area. Initially, it was surprising to observe how the Omicron variant was already so strongly present in our territory. Currently, Omicron is strongly represented in our country, and this finding explains the massive and prolonged diffusion of the virus [14]. In the present day, as in many other countries in the EU/EEA, we are observing an increasing diffusion of BA.5 as well as of other VOCs due to the attenuation of restraint measures [16,19,36]. Global and rapid diffusion of SARS-CoV-2 increases its ability to mutate, which represents a terrible application of the prediction of the Nobel Prize laureate, Joshua Lederberg, who defined the fight against microbes as our wits versus their genes.

## Figures and Tables

**Figure 1 viruses-15-00353-f001:**
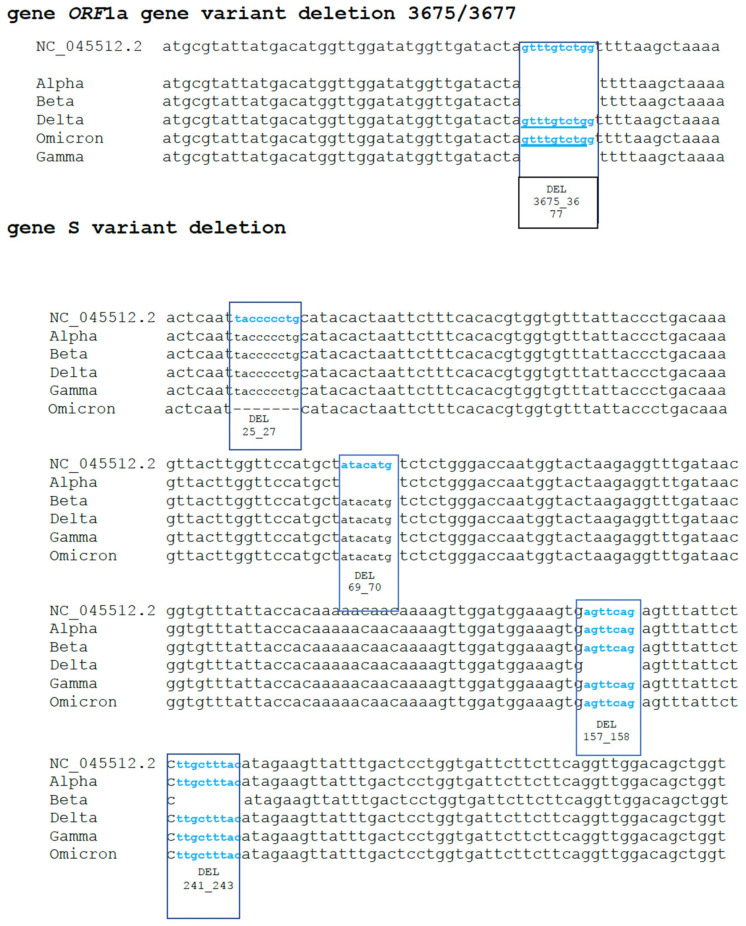
Portion of S gene sequence (of the SARS-CoV-2 genome): Wild type and variant showing the deletions 69–70 and portion of *ORF1a* gene with deletion 3675/3677.

**Figure 2 viruses-15-00353-f002:**
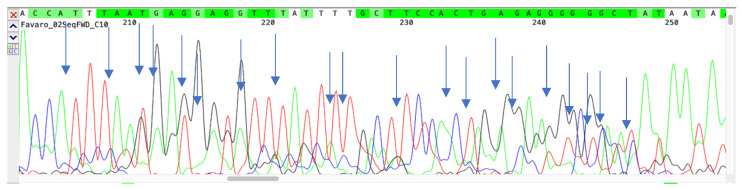
Electropherogram showing the sequence of samples in which more than one SARS-CoV-2 variant was present.

**Figure 3 viruses-15-00353-f003:**
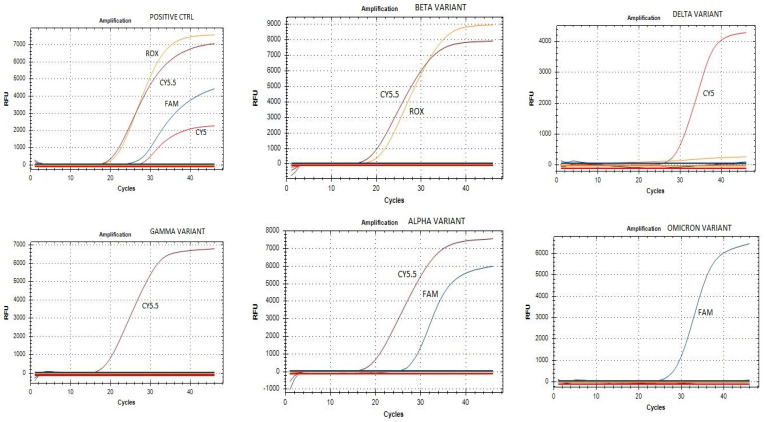
Specificity test of our assay.

**Figure 4 viruses-15-00353-f004:**
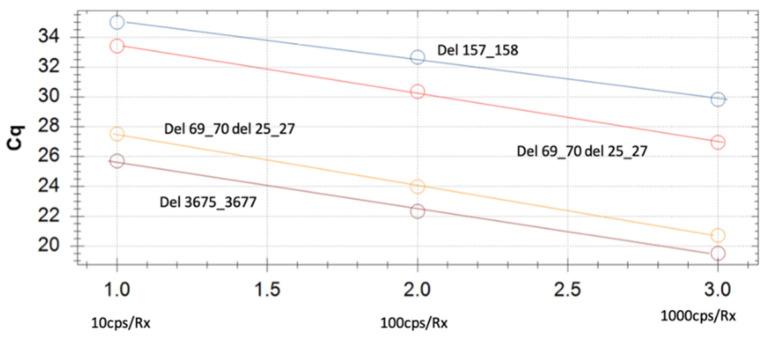
Sensitivity test of our assay.

**Table 1 viruses-15-00353-t001:** List of primers, probes, and their concentrations used in our assay.

Primers	Sequences	Concentration of Primer and Probe in Each Reaction
Forward D 69–70 (Alpha)	5′ GTT CCA TGC TMT CTC TGG G 3′	16 picomoli/µL
Reverse 69–70	5′ GTG GTA AAC ACC CAA AAA TG 3′	8 picomoli/µL
Forward D 25–27 (Omicron)	5′ AAC CAG AAC TCA ATC ATA CAC 3′	16 picomoli/µL
Reverse 25–27	5′ GTA TAG CAT GGA ACC AAG TA 3′	8 picomoli/µL
Forward D 241–243 (Beta)	5′ GGT TTC AAA CTT TAC ATA G 3′	4 picomoli/µL
Reverse 241–243	5′ ACC AGC TGT CCA ACC TGA AG 3′	2 picomoli/µL
Forward Δ 3675–3677(Alpha/Beta/Gamma)	5′ GGT TGA TAC TAG TTT GAA GC 3′	0.28 picomoli/µL
Reverse 3675–3677	5′ ACT CTC CTA GCA CCA TCA TCA 3′	0.28 picomoli/µL
Forward D 157–158 (Delta)	5′ AGT TGG ATG GAA AGT GGA GTT TAT	0.56 picomoli/µL
Reverse 157–158	5′ ACC CTG AGG GAG ATC ACG C	0.56 picomoli/µL
beta-actin F	5′ GAG GGT GAA CCC TGC AAA AG	2.5 picomoli/µL
beta-actin R	5′ CCC TCT AAG GCT GCT CAA TG	2.5 picomoli/µL
Probes	Labeling fluorophores	
Alpha/Beta/Gamma probe	5′ Cy5,5 TGC CTG CTA GTT GGG TGA TGC GT 3′ BHQ3	0.175 picomoli/µL
Alpha probe	5′ FAM TTG GTA CTA CTT TAG ATT CGA AGA3′BHQ 1	2.52 picomoli/µL
Delta probe	5′ Cy5 CTA GTG CGC CTA ATT GCA CTT TTGA 3′ BHQ3	0.28 picomoli/µL
Beta probe	5′ TxRed GTT ATT TGA CTC CTG GTG ATT 3′ BHQ2	2.0 picomoli/µL
Omicron probe	5′ FAM CAC ACG TGG TGT TTA TTA CCC TGA C 3′ BHQ1	4 picomoli/µL
beta-actin probe	5′ HEX GGT GGG GCA GTG GGG GCC ACC TTGT 3′ BHQ1	3 picomoli/µL

**Table 2 viruses-15-00353-t002:** Possible results of our assay and interpretation criteria for samples with a CT ≤ 38.

Fluorophores	FAM	ROX	Cy5	Cy5.5	HEX	Variant Detected
Interpretation	Δ 69/70 and 25/27	S Δ 241/243	S Δ 157/158	*ORF1a* Δ 3675–3677	IC	
Signals on each channel	POS	NEG	NEG	POS	POS	ALPHA
	NEG	POS	NEG	POS	POS	BETA
	NEG	NEG	NEG	POS	POS	GAMMA
	NEG	NEG	POS	NEG	POS	DELTA
	POS	NEG	NEG	NEG	POS	OMICRON
	NEG	NEG	NEG	NEG	POS	Wild Type or unknown variant

Table 2: An observed combination other than those described could indicate a new mutation or a coinfection of different variants.

## Data Availability

Not applicable.

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
