# Peer review of "Rapid System to Detect Variants of SARS-CoV-2 in Nasopharyngeal Swabs"

_viruses, 2023, doi:10.3390/v15020353_

Round 1

Reviewer 1 Report (New Reviewer)

 Favaro et al, conducted a study about the use of traditional real  time PCR as tool to discriminate between the VOC of SARS-CoV-2  virus.  The setting was to design and use of multiplex primers and probes as a mix in single tube for VOCs detection.

The authors suggest that the RT-PCR method as an alternative molecular system easy to  use and affordable  beside WGS to  discriminate between the VOCs.

WGS sure is expensive and more time consuming but it is very powerful in VOCs detection  while RT-PCR  will lose its power after emergence of new VOCs.

Therefore, I would suggest to the authors to  improve:

1.      The current state of knowledge (VOCs emergence)

2.      Correction in Line 118 :  8(8/400 ; 2%) not 1% error in calculation

3.      Clarifying  their obtained data by  showing

a.      Plots of RT-PCR particularly  when 2 VOCs are in the same NFS tube and whether the obtained data are corroborated with WGS  

4.      The 78 samples did not show any relevant mutation with respect to  wt strain is it based on sequencing or assay? 

Best regards

Author Response

Reviewer 1

Favaro et al, conducted a study about the use of traditional real  time PCR as tool to discriminate between the VOC of SARS-CoV-2  virus.  The setting was to design and use of multiplex primers and probes as a mix in single tube for VOCs detection.

The authors suggest that the RT-PCR method as an alternative molecular system easy to  use and affordable  beside WGS to  discriminate between the VOCs.

WGS sure is expensive and more time consuming but it is very powerful in VOCs detection  while RT-PCR  will lose its power after emergence of new VOCs.

Therefore, I would suggest to the authors to improve:

  1. The current state of knowledge (VOCs emergence) We have update the state of knowledge and added some references
  2. Correction in Line 118 :  8(8/400 ; 2%) not 1% error in calculation. Thank you for your suggestion. We have corrected
  3. Clarifying  their obtained data by  showing
  4. Plots of RT-PCR particularly when 2 VOCs are in the same NFS tube and whether the obtained data are corroborated with WGS. Unfortunately, in our study we did not performed WGS because it is not available in our center. However, Sanger methods has been commonly used in the identification of SARS-CoV-2 variants, particularly because their classification is based on S-gene sequence
  5. The 78 samples did not show any relevant mutation with respect to wt strain is it based on sequencing or assay? Thank you for your question, of course it has been based on both (Rt-PCR results as well as sequencing)

Reviewer 2 Report (New Reviewer)

Favaro et al. described a new rapid detection system for SARS-CoV-2 variant identification in the light of virus genetic variability and the impact of emerging VOC of pandemic management. The authors developed a TaqMan based Real-Time PCR assay discriminating virus variants based on the presence of specific deletion rather than point mutations, with a very high accuracy accordingly to reference procedures. Although interesting and very promising results were provided for the use of the proposed method in the diagnostic scenario, some important issues must be addressed. Within this context, my concerns are the following:

Major comments:

Technical issues are missing or poorly described along the manuscript. Since the aim of the study was the development and validation of an alternative diagnostic assay for virus variants identification, details should be provided. Moreover, the selection of clinical samples subjected to validation with the new assay is not clear. Mostly, the analysis was conducted to RNA samples stored for a long time of period (since March, 2021) and not re-tested for nucleic acid integrity. Along with the absence of any evidence about the performance of the new assay (just a brief mention about accuracy at lines 129-131), this strongly compromise the reliability of results. To allay this issue, a viral gene, such as the RdRp, should be targeted by specific probe as sample quality control rather that use the internal control (IC) for amplification only. Furthermore, the aim was the development of an easy system for VOC identification, but most of the work was addressed to distinguish between Alpha, Beta and Delta variants which are extinguished, while the characterization of Omicron sub-lineages was not improved. Indeed, the authors mentioned Omicron variants, without depicting which clade is detected or not, whether the sensitivity/specificity of the assay is comparable among the five Omicron sub-lineages or not, and other important aspects are missing. Moreover, in order to use the selected primers/probes panel as diagnostic tools, more experiments are required, such as clinical sensitivity and specificity, cross-reactivity, limit of detection. Hereafter, a detailed list of critical points to be addressed for further evaluation of the manuscript:

Lines 52-56: whole genome sequence is required if the full-length genome analysis is necessary. Variant classification could be performed based on Sanger sequence of specific portion of the Spike gene.

Lines 76-79: did you test any Omicron positive control? Please comment in the mentioned section.

Lines 80-83: I would suggest to re-write this part providing more details. Briefly, what was the Ct of selected samples? Did you select only high viral load samples or it was leave to chance?

Line 87: which internal control did you used? Please specify.

Line 92: “six primers and probes in a single tube” you already mentioned that. At this point, it’s better to detail the reaction condition, consisting of the amplification enzyme mix used (name of the kit and not the provider only), if the reaction was a one-step RT-PCR or a two-step system, and so on.

Lines 106-108: please map the primers location on the S gene and comment VOC-specific mutations detected. Moreover, specify if they were used for amplification reaction, for sequencing or for both analyses.

Lines 117-119: since you are testing the specificity of a new probe-based assay, this should be test on a similar number of different virus variants in order to get a significant result. Eight (for the Beta) and four (for the Delta) does not provide an accurate predictive value. Increase the number of these samples in order to assess the sensitivity/specificity of the assay for those variants.  Moreover, were samples with high, low or mixed viral loads? Please, provide at least a Ct range used for samples selection.

Lines 125-126: what’s the target of the FAM signal? It’s confusing and hard to understand since you did not mention haw the assay works.

Lines 116-128: please, re-write this part since it’s very confusing. Maybe you must state that all the samples selected were confirmed by sequencing, and results were/were not confirmed by your new assay. Clearly present the number of each variant screened and results of both diagnostic assays (real-time and Sanger).

Lines 129-131: you did not report any data concerning results of the new screening system but just the accuracy. I suggest to deeply describe this part since it is the main target of your work.

Figures 1: chromatogram are not very clear, please provide better resolution ones.

Figure 2: as stated above for figure 1, the chromatogram is not very well resolved. Thus, how could you exclude that double peaks are not due to noise?

Table 2: as commented above, which internal control did you use (the HEX probe)?

Lines 199-203: commercial test for variants identification are mostly based on annealing capability of specific probes targeting mutated regions in the S gene rather than the absence of S gene amplification. This led to the conclusion that, in case of co-infection, one, two or more probes bind or not to their specific targets. Conversely, if a re-infection is present, this does not affect the specificity of the assay at the sample collection time and no warry about which VOC was responsible for previous infection. Please, change and/or comment.

Minor comments:

Some parts of the manuscript are highlighted in yellow and light blue, what’s their meaning? Moreover, improve references.

Line 45: did you mean variant of interest (VOI), based on WHO classification, or the mentioned VOC? Please specify.

Line 61: “As BA.4 and BA.5”. Truncated sentence, please correct.

Line 63: “real time PCR probe”, I suggest to change in “TaqMan based real time PCR”.

Line 64: use characterise instead of detecting. In my opinion, this better fits to this context.

Line 73: nasopharyngeal should be abbreviated as NPS instead of NFS. Please correct though the manuscript.

Lines 115-116: which panel did you refer to? The commercial one?

Lines 182-184: the BA.5 is already widespread in Europe. Please update this statement.

Author Response

Reviewer 2

Favaro et al. described a new rapid detection system for SARS-CoV-2 variant identification in the light of virus genetic variability and the impact of emerging VOC of pandemic management. The authors developed a TaqMan based Real-Time PCR assay discriminating virus variants based on the presence of specific deletion rather than point mutations, with a very high accuracy accordingly to reference procedures. Although interesting and very promising results were provided for the use of the proposed method in the diagnostic scenario, some important issues must be addressed. Within this context, my concerns are the following:

Major comments:

Technical issues are missing or poorly described along the manuscript. Since the aim of the study was the development and validation of an alternative diagnostic assay for virus variants identification, details should be provided. Moreover, the selection of clinical samples subjected to validation with the new assay is not clear. Mostly, the analysis was conducted to RNA samples stored for a long time of period (since March, 2021) and not re-tested for nucleic acid integrity. Along with the absence of any evidence about the performance of the new assay (just a brief mention about accuracy at lines 129-131), this strongly compromise the reliability of results.

Thank you for your valuable comments. First of all we have added details in the methods section. Of course we have retested the stored nucleic acid from NSFs in order to be sure that the quality/integrity of RNAs was the same of the first amplification. We have not included this information in the first version trying to be concise and to provide the essential information. Moreover we would like to underline that the internal control of our assay has been chosen being an endogenous gene, which allow us to control not only the amplification but also the integrity of nucleic acid

To allay this issue, a viral gene, such as the RdRp, should be targeted by specific probe as sample quality control rather that use the internal control (IC) for amplification only.

We apologize but on this point we do not agree with your comment because the suggested gene is in itself the subject of mutations involved in the variants. It would not be possible to attribute its positivity and / or negativity to the presence / absence of mutation or alteration of the nucleic acid.

Furthermore, the aim was the development of an easy system for VOC identification, but most of the work was addressed to distinguish between Alpha, Beta and Delta variants which are extinguished, while the characterization of Omicron sub-lineages was not improved. Indeed, the authors mentioned Omicron variants, without depicting which clade is detected or not, whether the sensitivity/specificity of the assay is comparable among the five Omicron sub-lineages or not, and other important aspects are missing. Moreover, in order to use the selected primers/probes panel as diagnostic tools, more experiments are required, such as clinical sensitivity and specificity, cross-reactivity, limit of detection.

Unfortunately we do not agree with the reviewer’s comment, because when we performed our evaluation alpha, beta and delta variants were strongly represented, but not the same was for omicron (it was emerging). Moreover, in our opinion it is not required to distinguish the sub-variants of omicron, because they all have in common the same point mutation and the same deletion 25/27. We want to underline that the aim of our assay is to reduce the sample to submit to WGS, not to specifically identify the sub-variant

We would like to add a brief summary of the history of our paper, just for your knowledge:  Our paper has had a long revision process, with a first round of referees that asked us some revisions (that we completely accept and made). So we are expected that our paper would be re-evaluated from the same reviewers, unfortunately this not happened. The Editor told us that the first referees did not respond to the second round of evaluation, so the Editor decided to restart the process. As you can easily understand we are a little bit surprised and discourages because in this long time probably the paper lost its appeal and novelty. So we kindly ask you to take into consideration this unfortunate situation in your evaluation.

Indeed, the authors mentioned the Omicron variants, without describing which clade is detected or not, whether the sensitivity / specificity of the test is comparable or not among the five Omicron underlines, and other important aspects are missing. Furthermore, to use the selected primer / probes panel as diagnostic tools, more experiments are needed, such as clinical sensitivity and specificity, cross-reactivity, limit of detection.

Reading this comment it is almost clear that the real goal of our study was not comprised, and we apologize for that. Probably is our fault, we have not sufficiently described our assay. The test we propose, in fact, is not a diagnostic tool helping in identification of patient positive or not to SARS-CoV-2 infection. The purpose of our assay is a rapid identification of SARS-CoV-2 variant and it has to be run on already known positive nasopharyngeal swab. Its purpose is to allow a general characterization of the type of variant of the SARS-CoV-2 presents in a sample. It does not intend to be a precise identification of the variant and sub-variant. It only serves to give a general framework. If further investigations are necessary for epidemiological purposes, it is necessary to use the NGS approach.

So when you affirm “more experiments are needed, such as clinical sensitivity and specificity, cross-reactivity, limit of detection” it is clear that the main purpose of the assay has been not understood, likely because we did not extensively described and underlined this aspect. Samples used in our assay were  “nucleic acid extracted” from nasopharyngeal swabs (NFSs). Those NFSs were already been evaluated by commercially available system, and they are already known to be positive for SARS-CoV-2. Therefore, it was not necessary to define the sensitivity, the specificity, nor the cross reactivity of our assay. We have not to establish if the a sample is positive for SARS-CoV-2, we already known that our samples were positive. The positivity of a NSF was a selection criterion to send a sample to a second level of evaluation: the identification of the variant using our RT-PCR  assay. The work flow implied for our assay is: a) the laboratory receive a NFS; b) the laboratory perform a commercially available RT-PCR assay detecting SARS-CoV-2; c) the laboratory retested nucleic acid extracted from the initial NFS by our assay in order to a general characterization of the virus variant.

In addition, the specificity of our assay was demonstrated through sequencing (Sanger sequencing) as well as by the result of the control  included in our evaluation.

below, a detailed list of critical points to be addressed for further evaluation of the manuscript:

Lines

52-56: whole genome sequence is required if the full-length genome analysis is necessary. Variant classification could be performed based on Sanger sequence of specific portion of the Spike gene. We agree in fact, we performed Sanger sequencing of specific portion of the Spike gene

Lines 76-79: did you test any Omicron positive control? Please comment in the mentioned section.

We have specified that when the test was set up, the controls for the omicron variant were not yet available. In the present version we have performed an additional experiment by including a specimen characterized by WGS and identified as Omicron.

Lines 80-83: I would suggest to re-write this part providing more details. Briefly, what was the Ct of selected samples? Did you select only high viral load samples or it was leave to chance? Thank you for this comment we have added this information: NFSs whose CT were <= 38.

Line 87: which internal control did you used? Please specify. Thank you for your comment it was beta-actin gene (in the text was not reported but in the table 1 it was). We have added this information also in the text

Line 92: “six primers and probes in a single tube” you already mentioned that. At this point, it’s better to detail the reaction condition, consisting of the amplification enzyme mix used (name of the kit and not the provider only),  if the reaction was a one-step RT-PCR or a two-step system, and so on. We beg your pardon, honestly we do not understand this comment, all these information are already present in our paper: in the text (paragraph 2.3) as well as in the table 1

Lines 106-108: please map the primers location on the S gene and comment VOC-specific mutations detected. We beg your pardon but this information was already present in the table 1

Moreover, specify if they were used for amplification reaction, for sequencing or for both analyses.

This information is already present in paragraph 2.4

Lines 117-119: since you are testing the specificity of a new probe-based assay, this should be test on a similar number of different virus variants in order to get a significant result. Eight (for the Beta) and four (for the Delta) does not provide an accurate predictive value. This could be a good point. However at the time of our study those were what we had available. In addition the introduced the control from Twist Bioscience just to guarantee the quality of the results of our assay

Increase the number of these samples in order to assess the sensitivity/specificity of the assay for those variants.   Moreover, were samples with high, low or mixed viral loads? Please, provide at least a Ct range used for samples selection. We have introduced this information

Lines 125-126: what’s the target of the FAM signal? It’s confusing and hard to understand since you did not mention haw the assay works. We are afraid that for the reviewer it not clear the interpretation of the results of our assay. The latter has been reported in the table 2: FAM identifies both delection 69/70 and 25/27 and indicates the presence of Omicron. Moreover it has to be said that in a previous round of revision, a referee asked us to report data in such manner in the table 2

Lines 116-128: please, re-write this part since it’s very confusing. Maybe you must state that all the samples selected were confirmed by sequencing, and results were/were not confirmed by your new assay. Clearly present the number of each variant screened and results of both diagnostic assays (real-time and Sanger).

Thank you for your comment we have partially rewrite the paragraph

Lines 129-131: you did not report any data concerning results of the new screening system but just the accuracy. I suggest to deeply describe this part since it is the main target of your work. Actually in line 121_126 we reported the results observed. However the paragraph it has now partially rewritten (line 133-140)

Figures 1: chromatogram are not very clear, please provide better resolution ones. We are sorry about that. We have added a new version with a legend

Figure 2: as stated above for figure 1, the chromatogram is not very well resolved. Thus, how could you exclude that double peaks are not due to noise?

We have partially modified the figure, hoping that it is now suitable. Just our own consideration, and we beg your pardon, but we have up to 30 years experiences in sequencing so we are sure when we state that is not a noise.

Table 2: as commented above, which internal control did you use (the HEX probe)? The answer is yes, and you can find this information in probe list

Lines 199-203: commercial test for variants identification are mostly based on annealing capability of specific probes targeting mutated regions in the S gene rather than the absence of S gene amplification. This led to the conclusion that, in case of co-infection, one, two or more probes bind or not to their specific targets. Conversely, if a re-infection is present, this does not affect the specificity of the assay at the sample collection time and no warry about which VOC was responsible for previous infection. Please, change and/or comment.

We beg your pardon, honestly we are not sure of understanding the meaning. Our test is based on the presence of amplification, in one or more portions of the S gene or of the ORF1  gene which are specific for the presence of one or more deletions. In line 145-154 you can read that our test identifies directly the mutation if it is present.

Minor comments:

Some parts of the manuscript are highlighted in yellow and light blue, what’s their meaning? Moreover, improve references. The manuscript you read had already been corrected according to the indications of other referees. Some parts have been highlighted to make changes clear. In the present version we have omitted to highlight the text just to make more readable the paper (the paper has been deeply revised also in the English form)

Line 45: did you mean variant of interest (VOI), based on WHO classification, or the mentioned VOC? Please specify. Thank you for your suggestion, corrected

Line 61: “As BA.4 and BA.5”. Truncated sentence, please correct. Thank you for your suggestion, corrected

Line 63: “real time PCR probe”, I suggest to change in “TaqMan based real time PCR”. Thank you for your suggestion, corrected

Line 64: use characterise instead of detecting. In my opinion, this better fits to this context. We make the substitution

Line 73: nasopharyngeal should be abbreviated as NPS instead of NFS. Please correct though the manuscript Thank you for your suggestion, corrected

Lines 115-116: which panel did you refer to? The commercial one? We made the correction, panel referred to commercially available positive sample (used as control)

Lines 182-184: the BA.5 is already widespread in Europe. Please update this statement. We have update and added references

Reviewer 3 Report (New Reviewer)

Favaro et al. reported a RT-qPCR method for quick identification of SARS-CoV-2 variants. My major comments are that the introduction needs to be expanded and the manuscript needs extensive editing. 

Major comments: 

1. Lines 28–30, from the point of population genetics, SARS-CoV-2 evolution is driven by the large population size, high mutation rate (though RNA proofreading in SARS-CoV-2), and selection pressure. Discuss this. 

2. Lines 52–56, discuss the two WGS approaches—ARTIC amplicon and Nanopore. Make sure appropriate papers are cited. 

3. Line 58, here is the logic: 1). WGS is the standard, 2). WGS is time-consuming though, 3). variants can be lineage-specific and can be used for quick identification (cite the following two papers: Deng et al. 2020, 10.1126/science.abb9263, and Mei et al. 2021, doi.org/10.1093/molbev/msab265). Please write down this logic. 

4. In fact, I think the readers would appreciate it if the authors present a figure similar to Figure 3 in Mei et al. 2021, doi.org/10.1093/molbev/msab265. The new figure shows the deletions compared to the wild-type. It should be drawn on the weighted combination of all Sanger results. 

Minor comments: 

1. Line 10, I did not follow "-as well as to reference-". 

2. Line 12, "the SARS CoV-2 virus" → "the SARS-CoV-2 genome". 

3. Line 14, "25-27" → "25–27". Use en dash (–), not hyphen (-). Change this throughout the manuscript. 

4. Lines 12–15, "The primers and probes are designed to detect the main deletions that characterize the different variants, precisely in the S gene: deletions 25-27, deletions 69-70, 241-243 and 157-158 have been chosen."

→ 

"The primers and probes are designed to detect the main deletions that characterize the different variants. The amplification targets are deletions in the S gene: 25–27, 69–70, 241–243, and 157–158."

5. Line 15, "orf1a" → "ORF1a" in italic. 

6. Line 26, "several"? 

7. Line 37, "variant B1.351 and B.1.1.529 from South Africa" → "and variant B1.351 and B.1.1.529 from South Africa". 

8. Lines 43–45, "An even more … [12,13]." does not read right. Rewrite it and break it short. 

9. Lines 45–46, rewrite "It is debated …". 

10. Lines 85–86, remove "that we designed". 

11. Line 88, "Table1" → "Table 1". 

12. Lines 114–115, "Our PCR assay … expected." does not read right. 

13. Line 119, "All the remaining" → "All of the remaining". 

14. Line 134, "The latter probably" → "The latter was probably". 

15. Lines 168–170, rewrite "It is the … [25,26]".

Author Response

Reviewer 3

Favaro et al. reported a RT-qPCR method for quick identification of SARS-CoV-2 variants. My major comments are that the introduction needs to be expanded and the manuscript needs extensive editing. 

Major comments: 

  1. Lines 28–30, from the point of population genetics, SARS-CoV-2 evolution is driven by the large population size, high mutation rate (though RNA proofreading in SARS-CoV-2), and selection pressure. Thank you for your comment. We have added a sentence and a reference

Massimo Amicone, Vítor Borges, Maria João Alves, Joana Isidro, Líbia Zé-Zé, Sílvia Duarte, Luís Vieira, Raquel Guiomar, João Paulo Gomes, Isabel Gordo, Mutation rate of SARS-CoV-2 and emergence of mutators during experimental evolution, Evolution, Medicine, and Public Health, Volume 10, Issue 1, 2022, Pages 142–155, https://doi.org/10.1093/emph/eoac010

  1. Lines 52–56, discuss the two WGS approaches—ARTIC amplicon and Nanopore. Make sure appropriate papers are cited. We have added a reference, hoping to have correctly interpreted your suggestion
  2. Line 58, here is the logic: 1). WGS is the standard, 2). WGS is time-consuming though, 3). variants can be lineage-specific and can be used for quick identification (cite the following two papers: Deng et al. 2020, 10.1126/science.abb9263, and Mei et al. 2021, doi.org/10.1093/molbev/msab265). Please write down this logic. Thank you for your suggestion, we have rewritten the sentence and added the references
  3. In fact, I think the readers would appreciate it if the authors present a figure similar to Figure 3 in Mei et al. 2021, doi.org/10.1093/molbev/msab265. The new figure shows the deletions compared to the wild-type. It should be drawn on the weighted combination of all Sanger results. Thank you for your suggestion the figure has been added

Minor comments: 

  1. Line 10, I did not follow "-as well as to reference-". We have changed the period
  2. Line 12, "the SARS CoV-2 virus" → "the SARS-CoV-2 genome". Thank you for your suggestion
  3. Line 14, "25-27" → "25–27". Use en dash (–), not hyphen (-). Change this throughout the manuscript. We have corrected
  4. Lines 12–15, "The primers and probes are designed to detect the main deletions that characterize the different variants, precisely in the S gene: deletions 25-27, deletions 69-70, 241-243 and 157-158 have been chosen."→ "The primers and probes are designed to detect the main deletions that characterize the different variants. The amplification targets are deletions in the S gene: 25–27, 69–70, 241–243, and 157–158." We have replaced the sentence according to your suggestion
  5. Line 15, "orf1a" → "ORF1a" in italic. We have changed
  6. Line 26, "several"? we have changed several with many
  7. Line 37, "variant B1.351 and B.1.1.529 from South Africa" → "and variant B1.351 and B.1.1.529 from South Africa". We have replaced the sentence as suggested
  8. Lines 43–45, "An even more … [12,13]." does not read right. Rewrite it and break it short. We have corrected the sentence
  9. Lines 45–46, rewrite "It is debated …". We have corrected the sentence
  10. Lines 85–86, remove "that we designed". We have corrected
  11. Line 88, "Table1" → "Table 1". We have corrected
  12. Lines 114–115, "Our PCR assay … expected." does not read right. We have corrected
  13. Line 119, "All the remaining" → "All of the remaining". We have corrected
  14. Line 134, "The latter probably" → "The latter was probably". We have corrected
  15. Lines 168–170, rewrite "It is the … [25,26]". We have corrected

Round 2

Reviewer 2 Report (New Reviewer)

I appreciate that Favaro et al. responded to my critiques and addressed many of the critical points, including a better presentation/discussion of results. The manuscript was significantly improved compared to the previous version. However, although interesting and very promising results were provided for the use of the proposed method in the SARS-CoV-2 variants detection, some aspects do not completely meet my opinion. Within this context, my concerns are the following:

1.           Although the Omicron variant was not plenty spread when the report was conducted, this variant represents the most important threat right now, thus experiments targeting these VOC were required in order to in the current SARS-CoV-2 scenario. I’m not in agreement with the author’s statement that a further identification of Omicron sub-lineages is not necessary. In my opinion, a better preliminary identification of Omicron sub-variants would help for a better selection of those samples to submit for WGS. Albeit all the Omicron sub-lineages share the same deletion (25-27), many specific mutations characterise the sub-variants, which strongly influence virus spread, vaccine protection, and other epidemiological/clinical aspects, as you stated at lines 166-171. Thus, an update of the proposed assay would be advantageous. Moreover, I’m not in agreement with the author’s comment regarding sensitivity and specificity aspects of the proposed method. Although the assay was not used as a first-line SARS-CoV-2 diagnosis, the specificity and sensitivity must be addressed. If the assay’s specificity was not evaluated (cross reaction of VOC’s specific probes on different virus lineages), the risk to submit for WGS samples supposed to be Omicron rather that Alpha (as an example) might be occurred. Did you test probes’ cross-reaction? This would significantly improve the importance of the study. Similarly, if the sensitivity was not evaluated first, the risk to spend time and reagents to test samples not suitable for the assay performance might negatively affects the laboratory practice.

2.           Lines 856-860: I’m not completely in agreement with authors. Most of the commercial assays includes more than one probes (3-4) spanning the S gene, along with the RdRp and IC detection. Thus, the risk to fall in a false negative result imply that all the probes fail in generating signal, which is really rare.

Minor comments

Line 858: S gene and not protein.

Author Response

I appreciate that Favaro et al. responded to my critiques and addressed many of the critical points, including a better presentation/discussion of results. The manuscript was significantly improved compared to the previous version. However, although interesting and very promising results were provided for the use of the proposed method in the SARS-CoV-2 variants detection, some aspects do not completely meet my opinion. Within this context, my concerns are the following:

  1. Although the Omicron variant was not plenty spread when the report was conducted, this variant represents the most important threat right now, thus experiments targeting these VOC were required in order to in the current SARS-CoV-2 scenario. I’m not in agreement with the author’s statement that a further identification of Omicron sub-lineages is not necessary. In my opinion, a better preliminary identification of Omicron sub-variants would help for a better selection of those samples to submit for WGS. Albeit all the Omicron sub-lineages share the same deletion (25-27), many specific mutations characterise the sub-variants, which strongly influence virus spread, vaccine protection, and other epidemiological/clinical aspects, as you stated at lines 166-171. Thus, an update of the proposed assay would be advantageous.

Thank you for your observation, we understand the purpose of your request, however we want to emphasize that our test was not born with the aim of highlighting all the possible variants of the virus. We hope you will agree with us that it is practically impossible to develop a test capable of doing this, especially given the mutation rate of this virus.It would be like chasing the virus. The test is intended as a screening test to identify the samples to be subjected to WGS, adding specific mutations to the test would mean studying another test based on the new point mutations that characterize the subvariants, but given the limited number of fluorophores available this would not be possible

Moreover, I’m not in agreement with the author’s comment regarding sensitivity and specificity aspects of the proposed method. Although the assay was not used as a first-line SARS-CoV-2 diagnosis, the specificity and sensitivity must be addressed. If the assay’s specificity was not evaluated (cross reaction of VOC’s specific probes on different virus lineages), the risk to submit for WGS samples supposed to be Omicron rather that Alpha (as an example) might be occurred. Did you test probes’ cross-reaction? This would significantly improve the importance of the study. Similarly, if the sensitivity was not evaluated first, the risk to spend time and reagents to test samples not suitable for the assay performance might negatively affects the laboratory practice. 

Thank you for your suggestion we have added a paragraph line 113-118 and a figures (3,4) in the result section line 167-169

  1. Lines 856-860: I’m not completely in agreement with authors. Most of the commercial assays includes more than one probes (3-4) spanning the S gene, along with the RdRp and IC detection. Thus, the risk to fall in a false negative result imply that all the probes fail in generating signal, which is really rare. 

We are really sorry, but we didn’t find line 856-860 in our paper. We will be happy to answer this last point as soon as we have an indication of the sentence to which the reviewer refers 

Minor comments

Line 858: S gene and not protein. Thank you for your suggestion, the correction has been made on the text

Reviewer 3 Report (New Reviewer)

Thank you for making the revisions. I have no further questions.

Author Response

Thank you for making the revisions. I have no further questions.

Thank you for your valuable revision

This manuscript is a resubmission of an earlier submission. The following is a list of the peer review reports and author responses from that submission.

Round 1

Reviewer 1 Report

The manuscript describes a way to genotype the SARS-CoV-2 virus using real-time PCR that can be performed quickly in virtually any clinical or research laboratory.  I have several minor comments and a few major ones:

Minor Comments:

1) The paper could use some editing for grammatical errors and English language styling.

2) Line 31 states there were four main variants and then you list five.

3) Line 70 uses the abbreviation NFWs, which is not defined in the manuscript.

4) Section 2.1.  State the time frame that the samples are from.

4) Sections 2.2 and 2.3.  It is hard to determine if you are setting up a series of six individual reactions for each sample or if all of the primers and probes go into one large multiplex reaction.

5) Section 2.4.  It is not clear what amplicon is being sequenced.  Is it a large fragment of the S gene that contains all of the mutation sites?  If so is this a separate reaction from the your actual genotyping assay?  This needs more clarity.

6) Combine tables 1 and 2.  You can easily add the final oligonucleotide concentration to table 1 and completely eliminate table 2.  I might consider adding an additional column to table 1 indicating, which reaction each oligonucleotide is part of.  It would make it a lot more clear for anyone who wants to use this assay.

Major Comments:

1) There is no discussion of accuracy of your method compared to sequencing.  The results section indicates that you detected deletion 69/70 in 79 samples by sequencing and yet you had 200 samples that were classified as omicron by real-time PCR.  Does that mean the real-time PCR test was positive for the 69/70 deletion 200 times, but really it should only have been positive 79 times?

2) If you are stating your assay can identify Gamma variants then you need to include at least a few samples to prove that.

3) In line 100 you conclude that all other samples were wild type.  Can you really make that conclusion?  Could they be different variants that your assay does not identify?  That might be better written as, "the remaining samples were either wild type SARS-CoV-2 strains or some variant other than alpha, beta, gamma, delta, or omicron".

Author Response

Reviewers 1

We  wish to thank the reviewer for his valuable reviewing and for his helpful suggestions. Below our point to point responce

Minor Comments:

Q1) The paper could use some editing for grammatical errors and English language styling.

A1 The paper has been revised and edited again

Q2) Line 31 states there were four main variants and then you list five.

A2. I agree there was a misprint. It has been corrected

Q3) Line 70 uses the abbreviation NFWs, which is not defined in the manuscript.

A3) I agree, there was a typing error

Q4 Section 2.1.State the time frame that the samples are from.

A4: We apologize for the lack of the data. In the present version it has been added

Q5 ) Sections 2.2 and 2.3.  It is hard to determine if you are setting up a series of six individual reactions for each sample or if all of the primers and probes go into one large multiplex reaction.  

A5 we add a description that could be of helpful in understanding that is a Multiplex PCR

Q6) Section 2.4.  It is not clear what amplicon is being sequenced.  Is it a large fragment of the S gene that contains all of the mutation sites?  If so is this a separate reaction from the your actual genotyping assay?  This needs more clarity. 

A6) Thank you for you comment. We add a sentence that we hope can clarify that is a separate reaction. The sequence refers to a large fragment of S gene

Q7) Combine tables 1 and 2.  You can easily add the final oligonucleotide concentration to table 1 and completely eliminate table 2.  I might consider adding an additional column to table 1 indicating, which reaction each oligonucleotide is part of.  It would make it a lot more clear for anyone who wants to use this assay.

A7. We have combined table1/2 and renumbered in the text

Major Comments:

Q8 There is no discussion of accuracy of your method compared to sequencing.  The results section indicates that you detected deletion 69/70 in 79 samples by sequencing and yet you had 200 samples that were classified as omicron by real-time PCR.  Does that mean the real-time PCR test was positive for the 69/70 deletion 200 times, but really it should only have been positive 79 times?

A8 We apologize for the missing data we have rewrite methods, results and we have added a comment in the discussion

Q9 If you are stating your assay can identify Gamma variants then you need to include at least a few samples to prove that.

A9 Unfortunately, in the setting of the examined samples we did not find any Gamma. However, we used to evaluate our RT PCR a commercially available positive control, which included Gamma. This positive control gave the expected result. We have added this information in the method, and results

Q10 In line 100 you conclude that all other samples were wild type.  Can you really make that conclusion?  Could they be different variants that your assay does not identify?  That might be better written as, "the remaining samples were either wild type SARS-CoV-2 strains or some variant other than alpha, beta, gamma, delta, or omicron".

A10 We strongly agree. We have  included this sentence

Reviewer 2 Report

The paper describes a multiplex PCR for a rapid assessment of variants of SARS-COV2. 

-What was sequenced to confirm "the nature of the variants identified" (100/101). Amplicons? S-Gene? WGS?

-There is no correlation between the sequencing results and the assay results. Do they coincide or not? 

-To conclude no-result PCRs as wild-type Wuhan strains per se is difficult in my opinion, as no validation data for the PCR with prev. known/identified samples are presented.

-The paper describes a multiplex PCR for a rapid assessment of variants of SARS-COV2. However, additional data for PCR validation is needed. LOD for different variants? Colorcompensations needed? Verification with known samples of SARS-CoV2 variants

-Sequencing results must be more cleary presented and be correlated with PCR results. 

Overall the paper provides an interesting method to rapidly asses variants of SARS-CoV2, however, it needs major revision of the results part and a generally more concise presentation of the data obtained.

Author Response

Reviewers 2

The paper describes a multiplex PCR for a rapid assessment of variants of SARS-COV2. 

Q1-What was sequenced to confirm "the nature of the variants identified" (100/101). Amplicons? S-Gene? WGS?

A1 We have included more detail in method and results that we hope can be helpful in understanding the experiment. We have amplified and sequenced a large portion of the gene S of SARS CoV-2

Q2-There is no correlation between the sequencing results and the assay results. Do they coincide or not? 

A2. We thank for the comment, We have added additional details in the result

Q3-To conclude no-result PCRs as wild-type Wuhan strains per se is difficult in my opinion, as no validation data for the PCR with prev. known/identified samples are presented.

A3 Thank you for your comment, again we have added the results of an additional experiment. We used commercially available positive control to validate our result. This was used at the time of the study but were not included in the first version. Line 69-73

Q4 The paper describes a multiplex PCR for a rapid assessment of variants of SARS-COV2. However, additional data for PCR validation is needed. LOD for different variants? Color compensations needed? Verification with known samples of SARS-CoV2 variants.

A4. We are not completely sure of having understood this comment. If you meaning if the sample have been selected in advance among positive samples, the answer is yes. All the specimens included in the study were positive NSFs, and our assay was designed to be run on positive samples on which we want to quickly establish the variants present

Q5-Sequencing results must be more clearly presented and be correlated with PCR results.

A5. Thank you for your comment. Data has been added

Q6 Overall the paper provides an interesting method to rapidly asses variants of SARS-CoV2, however, it needs major revision of the results part and a generally more concise presentation of the data obtained.

A6. We thank for you valuable revision. We have revised and partially rewrite method, results, also by combining table 1 and 2

Round 2

Reviewer 1 Report

The authors have incorporated many of the suggested changes made by both reviewers, which has clarified the paper and resolved some of the previous concerns.  I do still have several comments/concerns regarding the manuscript:

1)       One thing the reviewers still did not do, despite both reviewers commenting on it, was to incorporate the accuracy of the RT-PCR method compared to sequencing (e.g. “99% of specimens categorized as alpha variant by sequencing were categorized as alpha variant with the RT-PCR assay”).

2)       Lines 123-124 states, “compatible with deletions 69-70 (Delta variant)…”.  However, table one shows a primer named “Forward 69-70 (alpha)”.  Lines 137-138 also indicate that deletions 69-70 are present in alpha and omicron variants.  Should line 123-24 say "alpha variant"?

3)       Table 2 indicates that the FAM channel should be positive in alpha and omicron variants. However, table 1 shows that the alpha probe is found in the Texas red channel.  Table 2 indicates the Cy5 channel should be positive for delta variants, but table 1 indicates the delta probe has a FAM fluorophore.

4)       Line 125-26 says, “200 samples for which the deletions 69-70 were not present, but showing the FAM signal in RT-PCR, were classified as omicron variants.”  However, in line 137-38 it says, “deletion 69-70 is present in both alpha and omicron lineages.”  If deletion 69-70 is present in omicron lineages (according to line 137-38) then how could you classify the 200 samples that were positive for FAM, but showed no deletion 69-70 as omicron?

5)       In many places in the manuscript Beta variants are described as having deletion 241/243 (e.g. lines 139 and 141).  In table 2 the Rox channel is described as having deletions 242/244 and in the abstract it is described as 242-244.  Which is correct, why are there different formats (242-244 vs. 242/244 vs. 241/243)?